# Sesame (*Sesamum indicum* L.): A Comprehensive Review of Nutritional Value, Phytochemical Composition, Health Benefits, Development of Food, and Industrial Applications

**DOI:** 10.3390/nu14194079

**Published:** 2022-09-30

**Authors:** Panpan Wei, Fenglan Zhao, Zhen Wang, Qibao Wang, Xiaoyun Chai, Guige Hou, Qingguo Meng

**Affiliations:** 1Key Laboratory of Molecular Pharmacology and Drug Evaluation, Collaborative Innovation Center of Advanced Drug Delivery System and Biotech Drugs in Universities of Shandong, School of Pharmacy, Ministry of Education, Yantai University, Yantai 264005, China; 2School of Biological Science, Jining Medical University, Rizhao 276800, China; 3Department of Organic Chemistry, School of Pharmacy, Naval Medical University, Shanghai 200433, China; 4School of Pharmacy, Binzhou Medical University, Yantai 264003, China

**Keywords:** sesame, nutritional value, phytochemical composition, bioactivity, food use, sesamin

## Abstract

Sesame (*Sesamum indicum* L.), of the Pedaliaceae family, is one of the first oil crops used in humans. It is widely grown and has a mellow flavor and high nutritional value, making it very popular in the diet. Sesame seeds are rich in protein and lipids and have many health benefits. A number of in vitro and in vivo studies and clinical trials have found sesame seeds to be rich in lignan-like active ingredients. They have antioxidant, cholesterol reduction, blood lipid regulation, liver and kidney protection, cardiovascular system protection, anti-inflammatory, anti-tumor, and other effects, which have great benefits to human health. In addition, the aqueous extract of sesame has been shown to be safe for animals. As an important medicinal and edible homologous food, sesame is used in various aspects of daily life such as food, feed, and cosmetics. The health food applications of sesame are increasing. This paper reviews the progress of research on the nutritional value, chemical composition, pharmacological effects, and processing uses of sesame to support the further development of more functionalities of sesame.

## 1. Introduction

Sesame (*Sesamum indicum* L.) is one of the earliest human production and consumption oil crops in the family of Pedaliaceae [1], rape, soybean, and peanuts, known as China’s four major oil crops. First discovered in ancient sites in Pakistan, sesame is a long-established cultivated crop [2]. It is distributed in countries such as India, China, and Malaysia. Chinese people have used sesame seeds for more than 5000 years [1,3]. Globally, India, Sudan, Myanmar, China, and Tanzania are the major producers of sesame. In recent years, the production of sesame seeds in African countries has increased, and Tanzania has replaced India as the leading producer of sesame seeds. According to the Food and Agriculture Organization of the United Nations, the global production of sesame in 2017 was 5.899 million tons, of which 806,000 tons were produced in Tanzania and 733,000 tons in China [4].

Sesame is widely grown and popular because of its highly aromatic odor and mellow flavor. In people’s lives, sesame seeds are often used to make a variety of foods, such as sesame oil, sesame paste, or to decorate other foods. The scientific nutritional value status of sesame seeds was established on a legal level in 2002 when they were included in the list of medicinal and food ingredients published by the former Chinese Ministry of Health.

Among the reviews on sesame, only one detailed review on the phytochemistry and ethno-pharmacology of sesame has been published in recent years [5]. Other reviews are either about a certain chemical composition in the sesame seeds [6], on some of the pharmacological effects of sesame [7], or the technical aspects of its production [8]. This review will focus on not only the phytochemical and pharmacological properties of sesame but also the economic-phytological and nutritional value of sesame.

## 2. Botanical Description

Sesame, a genus of Sesamum, is a member of the Pedaliaceae family. According to the difference in germplasm color, sesame can be classified as white sesame, black sesame, and yellow sesame, among which black and white sesame are the more common and widely grown dominant species, as shown in Figure 1. Black sesame has strong growth ability, lodging resistance, and drought resistance, while white sesame has high oil content and good quality and has the largest planting area and distribution. For other variegated varieties such as yellow sesame, its plants are mostly branched. In general, the oil content decreases gradually as the color of the germplasm deepens [9].

### 2.1. Morphology

Sesame is an erect annual herb that grows 60–150 cm tall. The stem is hollow or has a white pith. The sesame leaves are 3–10 cm long, 2.5–4 cm wide, and rectangular or ovate in shape with a slightly hairy surface. They are borne singly or 2–3 together in the leaf axils. The calyx lobes of sesame are 5–8 mm long and 1.6–3.5 mm wide, lanceolate in shape, and have a pilose appearance. The corolla of sesame is 2.5–3 cm long in a tube shape about 1–1.5 cm in diameter. It is white, often with a purplish-red or yellow halo. The four stamens are hidden inside the flower, the ovary is superior, 4-loculed and pilose outside, and flowering occurs in late summer and early autumn. The sesame capsule is rectangular in shape, 2–3 cm in length, and 6–12 mm in diameter, with longitudinal ribs on the surface and microscopic hairs on the epidermis. Sesame seeds are black or white, the black ones are called black sesame and the white ones are called white sesame [9,10,11]. The different forms of sesame seeds during growth are shown in Figure 2.

### 2.2. Cultivation

Sesame is a short-day crop in the subtropics, a temperature-loving plant, and is mostly produced in central Asia and North Africa. According to the branching character of sesame, there are two main types in cultivation and production. One is the monopole type, which is usually unbranched, with short internodes, bearing two to three capsules per node, with a hard stem, generally late maturity, and suitable for dense planting. Another type of sesame is the branched type, which generally matures earlier. It is branched, has long internodes, bears mostly one capsule per node, and should not be planted too densely [12,13].

The sesame cultivation area in China can be up to 45° N, mainly in the Yellow River and the middle and lower reaches of the Yangtze River. According to statistics, the national sesame cultivation area is about 790,000 hectares and the production is about 580,000 tons [14].

Because of its upright stems and low shade area, sesame is often used to grow in a mixture with short-stemmed crops. For example, it is mixed or intercropped with sweet potatoes, peanuts, soybeans, and other crops [15]. Sesame is more drought-tolerant, while beans are more moisture tolerant. Sesame and beans mixed crop is beneficial to prevent drought and flooding [16]. The oil content of sesame seeds ranges from 37% to 63%, depending on the cultivar and the growing season [17]. The differences in oil content of different varieties of the crop are related to the different effects of ecological factors, which influence the composition of the seeds, especially differences in precipitation or rainfall and light or sunlight radiation. The oil content also affects the size and color of the seeds [18,19].

### 2.3. Karyotype

The normal natural growth of sesame is a monoecious plant. Sesame is a diploid, self-pollinated oilseed crop [20]. In 1961, Joshi identified 36 species in the genus Hemp according to the Kewensis Index, with *Sesamum indicum* being the only cultivated species. The number of satellite chromosomes in cultivated *Sesamum indicum* was three pairs, and the karyotype types included 2A and 3A.

Morinaga et al., first reported that the number of chromosomes of cultivated sesame was 2*n* = 26, and each chromosome was relatively similar in size [21]. The study of sesame chromosomes in China began in the 1980s. The chromosome numbers of sesame were observed by the traditional squash method. It was confirmed that the four representative varieties of cultivated sesame in China were of 2*n* = 26 type, while the wild one introduced from Zambia (*Sesamum*
*l**aciniatum* klein) was of 2*n* = 32 type [22]. 

## 3. Nutritional Components

Sesame seeds are rich in fat, protein, minerals, vitamins, and dietary fiber. Sesame oil, which is obtained through traditional oil production methods, is rich in unsaturated fatty acids, fat-soluble vitamins, amino acids, etc. Studies have found that sesame seeds contain 21.9% protein and 61.7% fat, and are rich in minerals such as Fe and Ca [23]. Sesame seeds are rich in nutrients and have the reputation of being an “all-purpose nutrient bank” and the “crown of eight grains” [24]. The content of the main nutrients in sesame seeds is shown in Table 1. The nutrient fractions in sesame seeds are shown in Table 2.

### 3.1. Protein

The protein in sesame is a complete protein, of which the ratio of essential amino acid content is very similar to that of the human body [29]. Sesame protein is rich in variety, mainly including globulin, clear protein, alcoholic protein, and glutenin, of which globulin has the highest content and alcoholic protein has the least [30,31]. Sesame meal, a by-product of sesame processing, also contains about 50% protein. In vitro *tro* protein digestibility of sesame protein isolate using pepsin pancreatin enzyme systems showed that the sesame protein isolate had digestibility of 89.57% [32]. The high value of the in vitro protein digestibility showed that sesame protein isolate could be used to enrich and act as a supplement in some food systems, especially in developing countries where protein deficiency is a major health challenge for children.

Relevant studies have shown that peptides are not only utilized by nutrients that promote growth and development but also are important for the regulation and health of the organism. In Asian countries, the health benefits of black sesame seeds are considered to be greater than those of white sesame seeds because of the different shades of the seed coat [33].

A genome-wide association study of sesame seed coat color by Cui et al., found that the protein content of sesame seeds increased as the color of the seed coat deepened [34]. One of the more intuitive aspects is that the protein content of black sesame seeds is higher than that of white sesame seeds. The four proteins that have been reported in sesame are albumin, globulin (*α* and *β*), prolamin, and glutelin fractions [17], which were detected in seed species and are listed in Table 2. Nineteen essential amino acids have been isolated and identified from roots, seeds, flowers, stems, and leaves. They are as follows: alanine, arginine, aspartic acid, cysteine, glutamic acid, glycine, histidine, isoleucine, leucine, lysine, methionine, phenylalanine, serine, threonine, tyrosine, valine, tryptophane, proline, and *γ*-aminobutyric acid [25]. The essential amino acids isolated from sesame seeds are listed in Table 2.

### 3.2. Lipids

The lipids in sesame are mainly found in the seeds and are an important component of sesame. Sesame has the highest oil content among the major oil crops, up to 45~57%, which is why it has been known as the “Queen of Oil” since ancient times [35]. Sesame oil is reported to contain 80% unsaturated fatty acids and a small amount of saturated fatty acids [36]. 

Linoleic acid and linolenic acid are unsaturated fatty acids, which are essential fatty acids, but cannot be synthesized in the body and therefore must be obtained through diet. Linoleic acid can participate in cholesterol metabolism, increase the toughness of vascular epithelial cells, and contribute to growth and development. Linolenic acid can promote lymphatic B-cell differentiation and proliferation, and improve the acquired exogenous immunity [37]. Sesame oil, as an important product form of sesame, contains major unsaturated fatty acids, oleic (18:1) and linoleic (18:2) acids in the range of 26.60% to 54.85%, while minor unsaturated fatty acids are in the range of 0.13% to 0.89%. The content of saturated fatty acids varies from 0 to 10.58%, which is certainly a good supplement for essential fatty acids [33].

The relationship between the oil content of sesame seeds and protein content was reported by C. Li et al. [38]. The analysis of the association between oil content and protein content of sesame seeds using 112 polymorphic SSR markers revealed a strong negative correlation between them, and both were significantly influenced by genotype and environment (i.e., year and location).

The currently reported lipid, latifonin, was isolated and identified in sesame flowers [26]. The lipids isolated from sesame flowers are listed in Table 2. In addition, twelve unsaturated fatty acids have been found in sesame seeds. They are as follows: oleic acid, linoleic acid, palmitic acid, stearic acid, arachidic acid, linolenic acid, palmitoleic acid, lignoceric acid, caproic acid, behenic acid, myristic acid, and margaric acid [9,27]. The unsaturated fatty acids isolated from sesame seeds are listed in Table 2. However, it is well known that some side effects may occur with excessive consumption of any food, and the same is true for sesame seeds. For example, sesame seeds are high in unsaturated fatty acids and excessive consumption of sesame seeds can cause the body to gain weight, which is not friendly to dieters. Excessive intake of sesame seeds can lead to gastrointestinal discomfort and cause endocrine disruption in the body, which may lead to an increased risk of cardiovascular disease [39]. Omega-3 fatty acids can inhibit platelet aggregation, increasing the risk of bleeding [40,41], and sesame seeds are rich in this ingredient. Omega-3 fatty acids are known to lower blood pressure, and excessive consumption of sesame seeds is also associated with a risk of severe hypotension [42]. In addition, sesame seeds contain some antinutrients, such as oxalic acid and phytic acid, which, in excess, can have certain effects on the body, such as affecting the digestion and absorption of mineral elements and proteins in the gut and increasing the risk of kidney stones [43].

### 3.3. Vitamins

Vitamins also occupy a certain proportion in the nutrient composition of sesame, of which vitamin E is the richest in sesame [5]. In particular, vitamin E can be present in black sesame seeds at levels up to 50.4 mg/100 g. Studies have shown that *γ*-tocopherol is the major form of vitamin E in sesame seeds, while there is relatively less *α*-tocopherol. In vitro experiments have shown that *γ*-tocopherol has a stronger antioxidant capacity than *α*-tocopherol, but the overall vitamin E has a stronger functional activity [17].

Vitamin A, thiamine, riboflavinniacin, pantothenic acid, folic acid, ascorbic acid, *α*-tocopherol, *β*-tocopherol, *γ*-tocopherol, *δ*-tocopherol, and tocotrienol, all the twelve vitamins have been reported to be isolated from sesame seeds [17,28]. The vitamins isolated from sesame seeds are listed in Table 2.

### 3.4. Carbohydrates

Sesame seed hull is the major by-product of sesame seed oil extraction. It is mainly composed of 70–80% carbohydrate polymers (including hemicelluloses, cellulose, and pectic polysaccharides) [44]. Seven carbohydrates have been found in the seeds: D-glucose, D-galactose, D-fructose, raffinose, stachyose, planteose, and sesamose [17]. The carbohydrates isolated from sesame seeds are listed in Table 2.

### 3.5. Mineral Elements

Sesame seeds have been reported to be a source of several minerals, such as K (525.9 mg/100 g), P (516 mg/100 g), Mg (349.9 mg/100 g), Na (15.28 mg/100 g), Fe (11.39 mg/100 g), Zn (8.87 mg/100 g), and Mn (3.46 mg/100 g) [45].

### 3.6. Antinutrients

Antinutrients are substances that disrupt or prevent the digestion and absorption of nutrients, negatively affecting the health and growth of animals [43]. The main antinutritional factors in sesame seeds are oxalic and phytic acids and small amounts of tannins [46]. Until now, the range values of oxalic acid in sesame meal have been rarely reported, and Farran et al., detected 13% and 1.12% of oxalic acid and phytic acid in sesame hulls, respectively [46].

According to relevant analysis, oxalic acid in sesame hulls can cause more than half of the calcium in sesame to exist in the form of calcium oxalate, so that it is not well digested and used by livestock and poultry animals, thus preventing the digestion and absorption of nutrients [47]. In most cases, cooking techniques significantly reduce soluble oxalate, and should therefore enhance mineral availability. Aside from cooking, pairing high-oxalate foods with calcium-rich foods may offset soluble oxalate absorption. A normal calcium diet (800–1000 mg/day) should be able to offset potential inhibitory effects from dietary oxalates [43].

Phytate, also known as phytic acid or myo-inositol hexaphosphate, is another commonly considered “anti-nutrient” widely distributed in amongst the plant kingdom. Structurally, phytate is made up of six phosphate groups, attached to an inositol ring, with the ability to bind up to 12 protons total. These phosphate groups act as strong chelators, readily binding to mineral cations, particularly Cu^2+^, Ca^2+^, Zn^2+^, and Fe^3+^ [48]. These complexes are insoluble at neutral pH (6–7) and cannot be digested by human enzymes, affecting the digestion and absorption of minerals in the animal intestine. In addition, phytic acid can also bind to proteins in the intestine to form calcium–magnesium phytate protein complexes, which cannot be digested by protein hydrolases, thereby reducing the utilization of proteins and minerals [49]. Processing techniques such as soaking, fermentation, sprouting, germinating, and cooking can significantly alter the phytic acid content of sesame seeds, resulting in improved mineral utilization. In addition, the addition of a certain amount of phytase to the actual production of sesame flour in livestock diets can also improve the digestive utilization of calcium, phosphorus, zinc, and other mineral nutrients [43].

## 4. Phytochemistry

In addition to being rich in nutrients, sesame also contains many important functional components such as sesamin, sesamolin, sesamol, sesaminol, sesamolin phenol, and other lignan-like active ingredients [50]. The content of each component in sesame varies depending on the extraction method and external growing conditions, e.g., hot-pressed sesame oil has a higher content of sesamol, sesamin, and total lignans than cold-pressed and refined sesame oil [51]. The lignan content in sesame can be affected by factors such as strain, genotype, growing location (soil and weather), and growing conditions (irrigation, fertilization, and harvest time) [52].

A variety of phytochemical compounds have been identified and isolated from sesame seeds, seed oils, and various plant organs, including lignans, polyphenols, phytosterols, phenols, aldehydes, anthraquinones, naphthoquinones, triterpenoids, and other organic compounds. Among them, the lignan components isolated from sesame and their analytical methods are shown in Table 2.

### 4.1. Lignans

Sesame lignans are the main active ingredients in sesame seeds and have strong antioxidant activity [53]. Epidemiological studies have shown that sesame lignans have beneficial effects in regulating blood lipids and improving liver function. These properties are also responsible for the oxidative stability of sesame oil [6]. 

Sesamin represents about 50% of the sesame lignans, with sesamolin, sesamol, and sesaminol accounting for a small proportion of the weight. The mean levels are 2.48 mg/g (range 1.11–9.41 mg/g) and 1.72 mg/g (range 0.20–3.35 mg/g) for sesamin and sesamolin [6]. During the pressing process, sesame seeds undergo denaturing transformations represented by sesamin and sesamolin, into unsaturated low polymeric structural substances such as sesamol, sesaminol, and sesamolin phenol [54].

It has been shown that the lignan content of sesame seeds is strongly related to seed coat color, with black sesame seeds having the highest sesamin, sesamol, and total lignans content, while white sesame seeds have a relatively low sesamin content. The range of total lignin content in yellow, black, brown, and white sesame seeds was 2.52~5.94, 3.56~12.76, 2.66~6.68, and 2.83~5.66 mg/g, respectively [55].

A total of twenty-six lignans have been identified and reported from the aerial organs and seeds, namely sesamin, sesamolin, sesamol, (*+*)-episesaminone, (*+*)-episesaminol 6-catecho, pinoresinol, (*−*)-pinoresinol 4-*O*-glucoside, (*+*)-pinoresinol di-*O*-*β*-D-glucopyranoside, glucopyranosyl-(1→6)]-*β*-D-glucopyranoside], sesaminol, (*+*)-sesaminol 2-*O*-*β*-D-glucoside, (*+*)-sesaminol diglucoside, (*+*)-sesaminol 2-*O*-*β*-D-glucosyl(1→2)-*O*-[*β*-D-glucosyl(1→6)]-*β*-D-glucoside, sesamolinol, (*+*)-sesamolinol 4′-*O*-*β*-D-glucoside, sesamolinol 4′-*O*-*β*-D-glucosyl (1→6)-*O*-*β*-D-glucoside, matairesinol, samin, sesangolin, and disaminyl ether [9,28]. The relevant chemical structures of the lignans are shown in Figure 3.

#### 4.1.1. Sesamin

Sesamin is one of the most abundant lignans in the composition of sesame seeds and has good physiological activity. Studies have found that sesamin has good antioxidant properties, cholesterol lowering, lipid metabolism regulation, blood pressure stabilization, and anti-tumor effects [3]. Sesamin is metabolized in the body mainly by the action of cytochrome P-450. The metabolites of sesamin exist in body fluids and tissues mainly as glucosinolates and sulfate-conjugated forms, the excretion of sesamin is via bile, urine and feces, and the elimination of sesamin is mainly achieved by metabolism [56].

The variation in the content of sesamin, an oil-soluble lignan, may be related to the variation in the variety of sesame, local climate, and soil type, ranging from 60.14 to 69.10 mg/100 g [57]. When the unroasted and pressed sesame oil was decolorized with acidic white clay, some of the sesamin was formed by isomerization to the stereoisomer episesamin of sesamin with a content of about 0.28% [58].

#### 4.1.2. Sesamolin

Sesamolin is the second most abundant lignan in sesame. Since sesamolin does not contain phenolic hydroxyl groups, its antioxidant effect in the body is much weaker than that of sesamol. However, under certain heating conditions, sesamolin can gradually convert to sesamol. Taking advantage of this conversion feature, the addition of sesamolin enhances the antioxidant properties of oils and fats under heating conditions [5].

#### 4.1.3. Sesamol

Sesamol is present in low levels in sesame lignin, but it is the main flavor component and quality stabilizer of sesame oil, with good antibacterial and antioxidant properties. Studies have shown that sesamol is stable under sunlight conditions and can be used simultaneously with food additives containing Zn^2+^ and Mg^2+^, but not with strong oxidants [59].

#### 4.1.4. Sesaminol

Sesaminol is an important fat-soluble lignan. It is found at low level in sesame seeds but demonstrates good antioxidant properties and thermal stability. Under acidic conditions, sesamolin can be readily converted to sesaminol [60].

## 5. Sesame Product Development

In recent years, with the improvement of people’s living standards and health consciousness, the consumption of sesame and its products is on the rise. In the international market, the demand for sesame seeds has surged, with major consumer countries such as Japan, Korea, Turkey, Egypt, the United States, and Israel. In China, sesame oil is irreplaceable in the catering and cooking industry, with about 45% of sesame production used for sesame oil processing, 22% for sesame paste processing, 22% for peeled sesame processing, 5% for baked goods, and 6% for other uses [61]. Some studies reporting the application of sesame are shown in Table 3.

### 5.1. Food Uses

Sesame is one of the most popular foods among our consumers and is extremely versatile. In food production, sesame is a raw material for the production of traditional Chinese foods. The traditional foods in the market, including sesame oil, sesame paste, sesame candies, sesame cakes and other baked goods, as well as sesame filled dumplings, can be bought. These foods are very popular among consumers. It can be said that the Chinese people are the best at using sesame to produce a variety of delicious foods. The variety of sesame products is the widest in the world.

#### 5.1.1. Sesame Oil

Sesame oil is an aromatic oil extracted from sesame seeds and is a traditional product from the primary processing of sesame seeds, which can be used as edible oil. Sesame oil is rich in linoleic and linolenic acids as well as high amounts of biologically active substances such as lignans, natural vitamin E, and phytosterols [83]. The quality and nutritional content of the oil obtained from sesame seeds by cold pressing is high. The main unsaturated fatty acid in sesame oil is linoleic acid (46.9%), followed by oleic acid (37.4%). These fatty acids are essential fatty acids, because they cannot be synthesized in the organism and must be obtained through the diet. In addition, sesame oil is rich in vitamin E, which is dominated by gamma-tocopherol (90.5%) [84].

According to the Chinese Food Composition Table (2015), the average content of unsaturated fatty acids in sesame oil is 74.59%, while the average content of unsaturated fatty acids in olive oil is about 80%, which are very close to each other. Sesame oil is richer in flavor substances than olive oil. Therefore, it is more in line with the traditional dietary habits of Chinese consumers. In addition, sesame oil is commercially available at a much lower price than olive oil, making it more cost-effective [74].

In addition to the traditional water substitution, pressing, leaching, and filtering methods, the processing of sesame oil includes supercritical CO_2_ extraction, subcritical low-temperature extraction, microwave-assisted extraction, hydro enzymatic, and alkaline extraction methods. Some experiments have shown that sesame oil has anti-inflammatory, anti-swelling, and emollient effects, maintains capillary patency, and promotes inflammatory skin repair [85].

#### 5.1.2. Sesame Meal

The extraction of sesame oil leads to the production of a defatted by-product, sesame meal. This residue can be ground into a powder and used in cooking, thus bringing added value to the food industry. It contains a balanced amino acid composition of proteins, dietary fiber, and important bioactive compounds with antioxidant activity and health-promoting effects, such as lignans, mainly sesamin triglucoside and sesamin diglucoside [86].

The analysis of pressed sesame seed cake showed a high fiber content. As a prebiotic, it has several benefits for the stimulation of the gastrointestinal microbiota [87]. The positive regulation of the microbiota beneficially affects the host organism by promoting normal digestive function and defense against pathogens. In addition, soluble fiber forms a sticky layer in the small intestine that contributes to increase satiety. As nutrients reach further into the distal colon, the large amounts of nutrients exposed in the intestinal wall are reduced, leading to appetite hormonal changes that may contribute to weight loss [88]. 

#### 5.1.3. Processed Foods Related to Sesame

In recent years, there has been more and more research on the process of sesame compounding products. While retaining the flavor of sesame itself, sesame is processed with other raw materials to improve the nutritional value of the product, which can meet the needs of different consumer groups. Most of the more common sesame compounding products combine sesame with beans, grains, nuts, fruits, and vegetables. Process studies combining black sesame with yams, red dates, soybeans, and peanuts to make different kinds of compound beverages are also well established [89]. These compounding products have greatly diversified the sesame product range and contributed to the development of the sesame consumer market.

Due to the high nutritional value and balanced nutritional composition of sesame protein isolate, the addition of sesame protein isolate to food products can improve the nutritional quality of wheat-based bakery products. Researchers added different levels of sesame protein isolate to wheat flour to improve the protein content of wheat flour muffins. The nutritional quality of the muffins was improved by the addition of sesame protein isolate. The muffin with 15% sesame protein isolate was considered to be the best by professional judges in all aspects of taste and color [62].

Sesame protein isolates extracted by aqueous solution technology can be used as food ingredients, especially as thickeners, binders, and ingredients for baked goods [32].

Sesame oil, when combined with other vegetable oils, produces a blend with a good balance of essential fatty acids that can be used to produce healthy fat products [64]. Chia oil and sesame oil are important sources of essential fatty acids. However, when used alone, their ratio of omega-3 to omega-6 does not meet nutritional recommendations. Mixing these two different oils can improve this ratio to achieve balanced nutrition. In addition, the mixture of kiwi and sesame oils has a very stable physicochemical profile and has good antioxidant properties [63].

Sesame oil is rich in polyunsaturated fatty acids and can be used to make margarine. In addition, a blend of palm stearin and sesame oil can produce trans-fat-free baking ghee, which can be used in all-purpose ghee, liquid bread ghee, and pie crust ghee [65].

In terms of their healthful effects, vegetable oils are an important part of the human diet [90]. The phenolic composition, quality characteristics, and potentially beneficial properties of sesame seed oil make sesame a good health product [66]. Reported activities of sesame lignans include modulation of fatty acid metabolites, inhibition of cholesterol absorption and biosynthesis, antioxidant and protective vitamin E effects, hypotensive effects, improvement of liver function related to alcohol metabolism, and anti-aging effects. These beneficial activities may enable the use of lignans in functional health foods [67].

### 5.2. Antioxidants

In daily life, the addition of antioxidant components to cooking oils is essential to slow down their oxidation. Refined sesame oil has antioxidant properties that extend its shelf life in the food industry. This is because high temperature roasting is a necessary step in the refining of sesame oil. The Maillard reaction of reducing sugars with free amino acids during high temperature roasting not only enriches the flavor of sesame oil, but also increases the antioxidant activity. The sesamin and sesaminol content of sesame oil is also changed during the roasting process. This is due to the thermal decomposition of sesamolin at higher temperature, and sesamolin can be converted into sesamol. Sesame oil (unroasted) showed a relatively low content of sesamol, and roasting caused a significant increase in sesamol. Sesamol has been found to exhibit antioxidant activity in the body, so roasted sesame oil is resistant to rancidity [66]. The antioxidant content in sesame seeds is added to other oils to provide an antioxidant effect. For example, the addition of extracts from the outer skin of sesame seeds to sunflower oil can enhance its oxidative stability [68].

### 5.3. Applied in Traditional Chinese Medicine

According to the Shennong Ben Cao Jing, sesame seeds are used to treat “injuries in the middle of deficiency, nourish the five internal organs, benefit energy, grow muscles, and fill the marrow”. According to theories related to Chinese medicine, black sesame can exert its great activity by tonifying the liver and kidneys, and moistening the bowels and laxatives. It can be used to treat hemorrhoids, dysentery, constipation, cough, amenorrhea, dysmenorrhea, ulcers, and hair loss. Sesame seeds are also used as a topical ointment, lactation agent, diuretic, tonic, and pain reliever. In TCM applications, sesame oil is a good vehicle for drugs and helps them to cross the skin barrier [24].

### 5.4. Application to Pharmaceuticals

Sesame seeds and oil have significant pharmacological benefits and health benefits for the whole body, especially the liver, kidneys, spleen, and stomach organs. Its high oil content not only lubricates the intestinal tract, but also nourishes the internal organs [91]. Sesame oil also can promote burn healing by relieving minor burns or sunburns and aiding in the healing process. Sesame oil can also be used as a solvent, an oil carrier for medications, a skin softener, and a natural UV protector [50].

Sesame oil used for massage has some analgesic effect. For example, massage with sesame oil as an adjunctive treatment can effectively reduce the severity of pain in patients with chemotherapy-induced phlebitis [70]. Other data suggest that sesame oil massage is also effective in relieving acute traumatic limb pain [69].

### 5.5. Feeds

Sesame meal, which contains more than 45% crude protein and is rich in a variety of essential amino acids, is a high-quality vegetable protein resource [92]. Sesame meal is cheap and often used for animal feed, but there are certain oxalic acid and phytic acids in sesame meal. These anti-nutritional factors can affect the growth and development of animals, so sesame meal needs to be microbially fermented before it is used for animal feed [72].

Sesame meal can replace a certain amount of soybean meal in animal feed [93]. For example, replacing soybean meal in broiler feed with fermented sesame meal can improve the nutritional value of broilers, thus improving their production performance and can be used as a protein source in broiler diets [71]. Experimental results of replacing soybean meal with sesame meal in the diet of lactating ewes showed higher intake and digestibility of ether extract and greater milk production in ewes. The milk production cost of ewes fed with sesame flour is reduced [73].

### 5.6. Fertilizer

Sesame meal contains about 5.9% nitrogen, 3.3% phosphoric acid, and 1.5% potassium oxide, making it a good fertilizer. Fermented sesame meal used as fertilizer for crops such as watermelon, strawberries, and grapes can significantly improve the quality of agricultural products, such as increasing the sugar, vitamin C content, and fiber of fruits. Its use as a tobacco fertilizer increases the value of tobacco production and leads to better quality tobacco leaves. In the process, it not only increases the number of bacteria, actinomycetes, and fungi in the soil, but also increases the organic matter content of the soil [74].

### 5.7. Fuels

In ancient traditional usage, the stalks of sesame were used as fuel. In modern times, there is also fuel prepared by mixing waste cooking oil from sesame seeds with sesame flour suitable for use as an alternative fuel, which not only reduces waste emissions but also avoids food safety issues to some extent [94]. Currently, the world’s energy demand is met through non-renewable resources, such as petrochemicals, natural gas, and coal [95]. However, as the world’s fossil energy resources become increasingly depleted, energy demand is increasing. Sesame oil is expected to replace mineral oil as an alternative fuel in the future [75].

However, alternative fuels must be technically feasible, economically competitive, environmentally acceptable, and readily available [96]. Ahmad et al. [76] prepared biodiesel by transesterification of sesame oil with methanol in the presence of NaOH as a catalyst with maximum yield of 92% at 60 °C. Fueling engines with sesame biodiesel and its blends appear to have equal performance in terms of fuel consumption, efficiency, and power output compared to mineral diesel.

### 5.8. Cosmetics

As early as 1987, the U.S. FDA established the safety and application of sesame oil in cosmetics. In Japan, sesame oil is used not only as a base for pharmaceutical ointments and a diluent for various injections, but also in cosmetics such as eye shadow creams, lipsticks, and moisturizers. Currently, as the understanding of sesame gradually deepens, there are more and more studies on its deep processing and utilization [24]. For example, from the flowers and stems of sesame, fragrances used in the manufacture of perfumes are obtained. The myristic acid in sesame seeds is often used as an ingredient in cosmetics [79].

### 5.9. Insecticide

Sesamin has both fungicide and insecticide properties and can be used as a synergist for pyrethroid insecticides [78]. This property has been successfully applied to children’s hair to prevent lice infestation. Sesame oil can also be used in combination with other substances as a wood protection agent to prevent tree damage by termites in the field [77].

### 5.10. Environmental Protection

Activated carbon was prepared from sesame oil cake by thermal method, sulfuric acid method, and zinc chloride method, which can effectively remove hexavalent chromium from chromium plating wastewater, and thus can achieve the purpose of environmental protection [80].

### 5.11. Bio-Use Aspects

Genetic engineering can complement traditional plant breeding methods by making precise changes to plants in a short period of time. The experimenters introduced diacylglycerol acyltransferase isolated from sesame into the soybean variety Dongnong 47 and used agrobacterium-mediated transformation to produce a soybean high-oil transgenic line more suitable for breeding [81]. It was found that the seed oil content of transgenic soybean overexpressing SiDGAT1 increased by more than 1.0 percentage points on average. In addition, transgenic plants expressing SiDGAT1 significantly reduced the protein and soluble sugar content in mature seeds. Therefore, engineering the SiDGAT1 enzyme is an effective strategy to improve the oil content and value of soybean.

### 5.12. Other Uses

Sesame is a very versatile plant that has many uses beyond those mentioned above. For example, hot pressed sesame oil can be used to make copy paper. The fumes from burning sesame oil can be used to make high grade ink. Sesame can also be used by industry to make lubricating soap [97].

Sesame degreasing powder can be used as a stabilizer for soybean oil-in-water (O/W) emulsions [82]. As the amount of defatted sesame powder increases, it results in a smaller droplet size and higher stability of the emulsion against agglomeration or emulsification. With the addition of 3.0% defatted sesame powder, the oil content increases within a certain range. This facilitates the formation of gel-like emulsions with smaller particle size and better emulsion stability.

## 6. Main Physiologically Active Effects

Several studies have reported that natural lignans present in sesame seeds, such as sesamin and sesamolin, have various pharmacological effects, which contain anti-inflammatory, antioxidant, anti-cancer, anti-hypertensive, anti-melanogenic, auditory protection, anti-cholesterol, and other strong bioactive effects. They also have a protective effect on the heart, liver, and kidneys [98,99,100,101]. The health functions and related mechanisms of sesame are listed in Table 4.

### 6.1. Antioxidant Effect

Oxidative stress is the excessive production of free radicals that disrupt the homeostasis of the antioxidant balance. The production and removal of free radicals are in equilibrium, and this equilibrium is maintained by the mechanism of “redox” [118].

Sesamin has been found to scavenge free radicals and exhibit antioxidant activity in the body. Ruankham et al. [102] found that sesamin inhibited H_2_O_2_-induced reactive oxygen species (ROS) production in human neuroblastoma and increased catalase (CAT) and superoxide dismutase (SOD) activities to protect cells from oxidative stress. H_2_O_2_ was also found to reduce SIRT1 and SIRT3 levels, but sesamin was able to reverse this alteration. It is hypothesized that sesamin can affect the SIRT1-SIRT3-FoxO3a signaling axis and attenuate H_2_O_2_-induced oxidative damage. It was also found that increased apoptosis after H_2_O_2_ treatment was associated with cystatin-3/7 activation, BAX upregulation, and BCL-2 downregulation, and sesamin was able to reverse these changes to reduce the level of apoptosis and could be used as an anti-apoptotic drug. 

Similarly, Fan et al. [103] identified SIRT3 as a target of sesamin inhibition, capable of normalizing cardiac SIRT3 and SOD induced by aortic constriction surgery in mice, and blocking cardiac remodeling dependent on SIRT3 by reducing ROS levels.

### 6.2. Cholesterol-Lowering and Lipid-Regulating Effects

Various studies have shown that sesamin has potent lipid-lowering properties. Sesamin’s lipid-lowering effects are mainly attributed to its ability to affect key steps in fatty acid and cholesterol metabolism and to reduce atherogenesis-triggering LDL, VLDL, and TG levels, as well as to increase atheroprotective HDL levels.

Liang et al. [104] showed that sesamin is beneficial in regulating lipids and lowering cholesterol. Sesamin dose-dependently down-regulated the mRNA of NPC1L1, ACAT2, MTP, ABCG5, and ABCG8, and had effects on the expression of genes involved in cholesterol absorption-related proteins and enzymes. Sesamin had no effect on LDL-C receptor mRNA and hepatic SREBP2. However, it caused a dose-dependent increase in mRNA for CYP7A1, and a dose-dependent decrease in mRNA for HMG-CoA reductase and LXR*α*. This suggests that cholesterol lowering by sesamin is associated with down-regulation of sterol transporter genes related to cholesterol absorption. 

Sesamin affects aspects of cholesterol metabolism. It decreases cholesterol synthesis and CYP7A1 expression and activity by downregulating HMGCR and can cause downregulation of many different sterol transporter levels, leading to a decrease in cholesterol absorption and a corresponding increase in fecal excretion of neutral steroids. Sesamin also contributes to the maintenance of homeostatic cholesterol levels of ABCG1 and ABCA1 through upregulation of RCT sterol transporter expression as well as through upregulation of PPAR*γ*1, LXR*α*, and MAPK regulatory pathways [119]. Therefore, sesamin may be a promising agent for reducing LDL and VLDL levels and increasing HDL levels.

### 6.3. Protect Liver and Kidney Function

It was found that sesamin significantly reversed the elevation of ALT, AST, and total bilirubin, induced the elevation of SOD and GSH-Px antioxidant activities, and significantly reduced the elevation of IL-6 and COX-2 in liver fibrosis mice [100]. Sesamin can significantly inhibit the activity of NF-κB and prevent its transfer from cytoplasmic to nuclear components, which has good hepatoprotective and anti-fibrotic effects.

Guo et al. [101] found that sesamin could reduce serum ALT, AST, ALP, urea nitrogen, and creatinine levels in adriamycin-induced rats to different degrees, which could effectively protect liver and kidney functions. Meanwhile, sesamin could significantly reduce MDA and 4-hydroxynonenal content in adriamycin-induced liver and kidney tissues, and increase the activities of antioxidant enzymes SOD, CAT, and GPX in liver and kidney tissues. It is suggested that sesamin can alleviate adriamycin-induced hepatorenal toxicity by inhibiting oxidative stress.

Rousta et al. [105] found that different concentrations of sesamin had a protective effect on LPS-induced acute kidney injury in mice. This significantly reduced serum urea nitrogen and creatinine levels, and reduced the increases in MDA, SOD activity, catalase activity, glutathione content, and Nrf2 levels in LPS-induced renal tissues, but had no significant effect on nitrite content. Meanwhile, sesamin was able to normalize the abnormalities of NF-κB, Toll-like receptor 4, COX-2, DNA breakage, TNF-*α*, and IL-6 levels produced by LPS induction, suggesting that sesamin may counteract LPS-induced acute kidney injury by reducing renal oxidative stress, inflammation, and apoptosis.

A study by Cao et al. [106] showed that sesamin had a significant dose-dependent decrease in fluorine-induced kidney injury and apoptosis in carp, and was able to inhibit renal ROS production and suppress oxidative stress. It showed that this had a significant inhibitory effect on renal cystatin-3 activity and reduced the level of p-JNK protein in the kidney of a fluorine-exposed group of fish, indicating that sesamin protects against renal oxidative stress and apoptosis through the JNK signaling pathway.

### 6.4. Anti-Inflammatory Effect

Studies have shown that sesamin has anti-inflammatory effects [107]. TNF-*α* is known to play an important role in the formation of rheumatoid arthritis [108]. Khansai et al., found that sesamin significantly reduced the mRNA expression of IL-6 and IL-1 in human primary synovial fibroblast cell lines, indicating that sesamin inhibited TNF-*α*-induced pro-inflammatory cytokine mRNA expression [109]. Sesamin catechol conjugates are thought to be the major metabolites of sesamin present in human plasma following oral administration of sesamin. Catechol glucuronides exert anti-inflammatory effects through demyelination in macrophage-like J774.1 cells, thereby inhibiting the expression of interferon beta and inducible nitric oxide synthase. It was found that SC1, one of the sesamin metabolites of CYP450, had stronger anti-inflammatory activity than sesamin itself in murine macrophage-like J774.1 cells [110].

### 6.5. Hypoglycemic Effect

Type 2 diabetes mellitus, a chronic metabolic disorder characterized by altered fat, protein, and carbohydrate metabolism, is considered to be a global public health problem with increasing prevalence worldwide. Experimental studies have shown that white sesame oil can help reduce the harmful effects of diabetes [111]. The researchers randomized male Sprague Dawley rats into a standard diet group, a normal control group, and a diabetic control group, as well as a diabetic sesame oil group, which was fed a diet containing 12% white sesame oil, and blood samples were analyzed at 0, 30, and 60 days. Differences between groups and between days were assessed using two-way repeated measures ANOVA. In the beginning, fasting blood-glucose and insulin were similar in the two diabetic groups, with a mean of 248.4 ± 2.8 mg/dL and a mean of 23.4 ± 0.4 μU/mL, respectively. After 60 days, fasting blood-glucose was significantly higher in the diabetic control rats (298.0 ± 2.3 mg/dL) than in the diabetic sesame oil group (202.1 ± 1.0 mg/dL) (*p* < 0.05). The results showed that consumption of white sesame oil significantly reduced hyperglycemia and other biomarkers of liver stress, as well as protecting heart and kidney health.

In an 8-week open-label randomized dietary intervention study, Devarajan et al., found significantly lower fasting and postprandial glucose at weeks 4 and 8 in patients with type 2 diabetes treated with a sesame oil mixture, glibenclamide, or a combination of glibenclamide and sesame oil mixture (*p* < 0.001) [112]. HbA1c, total cholesterol, triglycerides, LDL cholesterol, and non-HDL cholesterol were significantly lower (*p* < 0.001), whereas HDL cholesterol increased significantly at week 8 (*p* < 0.001). This result also demonstrated that sesame oil can reduce the symptoms of hyperglycemia in type 2 diabetic patients and improve lipid profile.

### 6.6. Protection of the Cardiovascular System

Hypertension is a risk factor for cardiovascular disease. It is estimated that the number of people with hypertension will exceed 1.5 billion worldwide by 2025. Studies have shown that a diet rich in polyunsaturated fatty acids and vitamin E would be beneficial in reducing hypertension and cardiovascular morbidity. Sesame seeds are rich in polyunsaturated fatty acids, phytosterols, lignans, and vitamin E, which have a beneficial effect on blood pressure [113,120]. 

A clinical study by Helli et al. [114] found that sesamin supplementation significantly reduced serum MDA levels and increased TAC levels in patients with rheumatoid arthritis. Patients’ body weight, BMI, and systolic blood pressure also decreased significantly, suggesting that sesamin may reduce cardiovascular risk factors in patients with rheumatoid arthritis.

The results of the analysis by Khosravi-Boroujeni et al. [121] showed that the consumption of sesame seeds helped to lower blood pressure, the magnitude of which needs to be further studied. A study by Li et al. [122] found that sesamin reduced heart weight, left ventricular weight, cardiomyocyte size, and left ventricular weight/body weight to heart weight/body weight ratio, significantly reduced mitochondrial and myofiber damage, suppressed systolic blood pressure elevation, and improved cardiac damage in spontaneously hypertensive rats. At the same time, sesamin increased T-AOC, decreased cardiac MDA content and nitrotyrosine levels, and inhibited TGF-*β*1 protein levels and mRNA expression in the left ventricle. 

Thuy et al. [115] showed that oral administration of sesamin for 4 weeks significantly improved changes in heart rate, blood pressure, and QT interval in patients with streptozotocin-induced type I diabetes. The above findings demonstrate the potential protective function of sesamin on the cardiovascular system.

### 6.7. Anti-Tumor Effect

Several studies have shown that sesamin has potent anticancer properties. The anticancer effects of sesamin are mainly attributed to its anti-proliferative, pro-apoptotic, anti-inflammatory, anti-metastatic, anti-angiogenic, and pro-autophagocytic activities. Although the exact signaling events triggered by sesamin in cancer cells have not been fully revealed, STAT3, JNK, ERK1/2, p38 MAPK, PI3K/AKT, caspase-3, and p53 signaling pathways play a key role in mediating the anticancer effects of sesamin [3].

Cavuturu et al. [123] showed that sesamin competitively inhibits the formation of the *β*-catenin/Tcf4 complex, blocking the typical Wnt signaling pathway associated with colon cancer. Sesamin has been shown to be a potent anti-cancer agent, which can effectively slow down the development and progression of tumors.

### 6.8. Other Aspects

Studies have confirmed the inhibitory effect of sesamin on osteoclastogenesis in humans. The effect of sesamin on human osteoblasts was investigated in terms of differentiation and function of human PBMCs induced by M-CSF and RANKLin by Wanachewin et al. [116]. Treatment with sesamin significantly reduced the number of differentiated osteoblasts observed by TRAP staining. Sesamin did not reduce NFATc1 gene expression, which was in contrast to the decreasing trend of CathK and TRAP expression. DC-STAMP, but not Atp6v0d2, was also significantly decreased in the presence of 14lM sesamin. Sesamin may inhibit human osteoclast differentiation, precursor cell recruitment, and F-actin formation [116]. The reduction in the area of resorption pits and the release of collagen from bone fragments under sesamin treatment reinforce the inhibitory effect of sesamin on the differentiation and function of osteoclasts. Sesamin is a promising phytochemical agent that inhibits osteoclast differentiation and function. 

Sesame oil and sesamin have been shown to have an ameliorative effect on hearing loss and to increase the activity of hair cell damage. It has been shown that sesame oil and sesamin have a protective effect on hearing, and this effect is achieved by altering the expression of genes associated with hearing loss. The experimenters studied the auditory-associated gene Tecta using a 3-(4,5-dimethylthiazol-2-yl)-2,5-diphenyltetrazoliummi (MTT) assay [117]. Treatment of ototoxic zebrafish larvae with sesame oil and sesamin induced proliferation of auditory cells. Sesame oil induced a significant increase in the number of kinocilia, and neural ast recovery in a zebrafish sensory nerve injury model indicated that sesame oil and sesamin enhanced their activities. Tecta expression in the tip region of cochlear hair cells increased with their growth. Researchers have also demonstrated the effectiveness of sesame oil in a mouse model of NIHL. Mice treated with sesame oil reduced their hearing threshold, implying that sesame oil has an ameliorative effect on NIHL disease. Thus, these data clearly indicate that sesame oil and sesamin are candidates for the treatment and recovery of hearing loss.

## 7. Safety

Against the backdrop of an accelerating aging society, an increase in diseases caused by lifestyle habits, and rising medical costs, the number of people consuming nutritional functional foods, specific health foods, supplement for health promotion, disease prevention, and treatment is increasing. When food and drugs are used together, it is possible to enhance or diminish the effectiveness of the drug. Because some food factors can affect drug metabolism, it is critical to clarify drug–sesamin interactions to clarify safety issues.

In vivo studies using rats were conducted by Sakaki et al. [124]. They observed the course of plasma concentrations of diclofenac over time in rats given sesamin for 3 days after administration of diclofenac, a nonsteroidal anti-inflammatory drug. They found no significant differences in C-max, T-max, and AUC (0–24 h) of diclofenac between the sesamin group and the non-sesamin group. Based on this result, we can conclude that people taking sesamin supplements at standard doses do not experience significant interactions with the drug.

Sesame seed oil was given to ten mice at different doses of 0.5, 1.0, 1.5, 2.0, and 3.0 g/kg, orally. The control group received saline. For 48 h, the groups will be observed, and mortality will be recorded at the end. At the end of 48 h, no toxicity was shown by sesame seeds oil [125]. 

Different doses in the range of 50–2000 mg/kg body weight were given to rats to check the toxicity of the ethanolic extract of sesame seeds. No side effects were observed even after 7 days of observation [126]. Usual signs of toxicity, including convulsions, piloerection, diarrhea, and ataxia, were not detected. There was no alteration in the motor activity such as corneal reflex, number of pats, body tone, and respiration.

## 8. Conclusions and Perspectives

Sesame, one of the oldest oil crops available, is a versatile plant that contains a high nutritional value. In addition to making oil, sesame seeds are often made into sesame paste, sesame milk, sesame paste, and other foods. Currently, with the continuous research on sesame, more bioactive components are being explored and applied, effectively promoting the development of the sesame processing industry. People have shown great interest in exploring this high economic value and nutritious crop. 

More than 180 phytochemical components of sesame seeds, seed oils, and various organs have been isolated and identified through modern research, including lignans, polyphenols, phytosterols, phenols, anthraquinones, cerebrosides, fatty acids, vitamins, proteins, essential amino acids, and sugars. Numerous studies have shown that sesame seeds and sesame oil are richer in phytochemicals and have higher nutritional value than other plant organs of sesame. Sesamin, sesamol and other chemical components have a variety of pharmacological effects and are of great benefit to human health, and can be used in the treatment of diseases such as anti-inflammatory, antioxidant, anti-cancer, anti-melanogenic, auditory protection, anti-cholesterol, and anti-aging, and have a protective effect on the heart, liver, and kidneys.

There are few studies on the genetics of sesame as well as on the improvement of genetic variation and root system. In-depth research on sesame is still necessary to further increase the yield and quality of sesame and to improve the related traits of sesame.

In addition, relatively few studies have been conducted on the drug conformation, pharmacokinetic, and bioavailability of sesame bioactive substances. Some experimental studies lack basic pharmacological parameters such as negative or positive controls, maximum and minimum dose responses, and temporal responses. These lead to difficulties in sustainability and reproducibility of the data. Therefore, further research is needed to explore the pharmacological effects of sesame.

Sesame is drought tolerant and easy to grow, and is suitable for intercrop rotation cultivation, making it a very promising crop. In drought-prone regions such as Africa, increasing the cultivation of drought-resistant crops such as sesame can create jobs and increase the income of local people. With the continuous progress of modern technology, the production scale of sesame in the world is expected to be further expanded, thus creating higher economic benefits.

## Figures and Tables

**Figure 1 nutrients-14-04079-f001:**
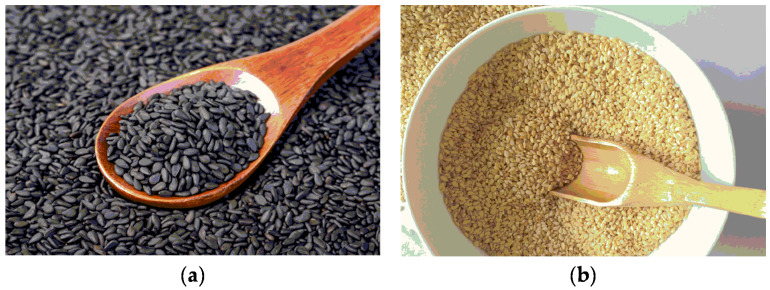
Sesame seeds of different colors. (**a**) Black sesame; (**b**) white sesame.

**Figure 2 nutrients-14-04079-f002:**
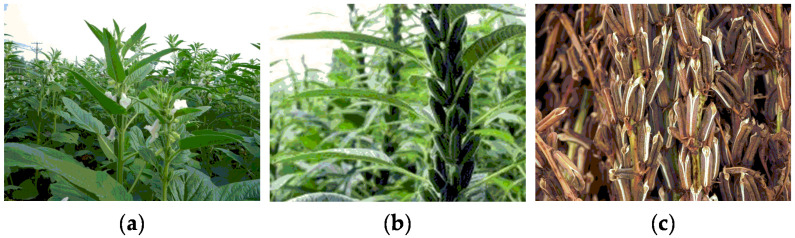
Different growth states of sesame. (**a**) Sesame blooming; (**b**) unripe sesame pods; (**c**) ripe sesame pods.

**Figure 3 nutrients-14-04079-f003:**
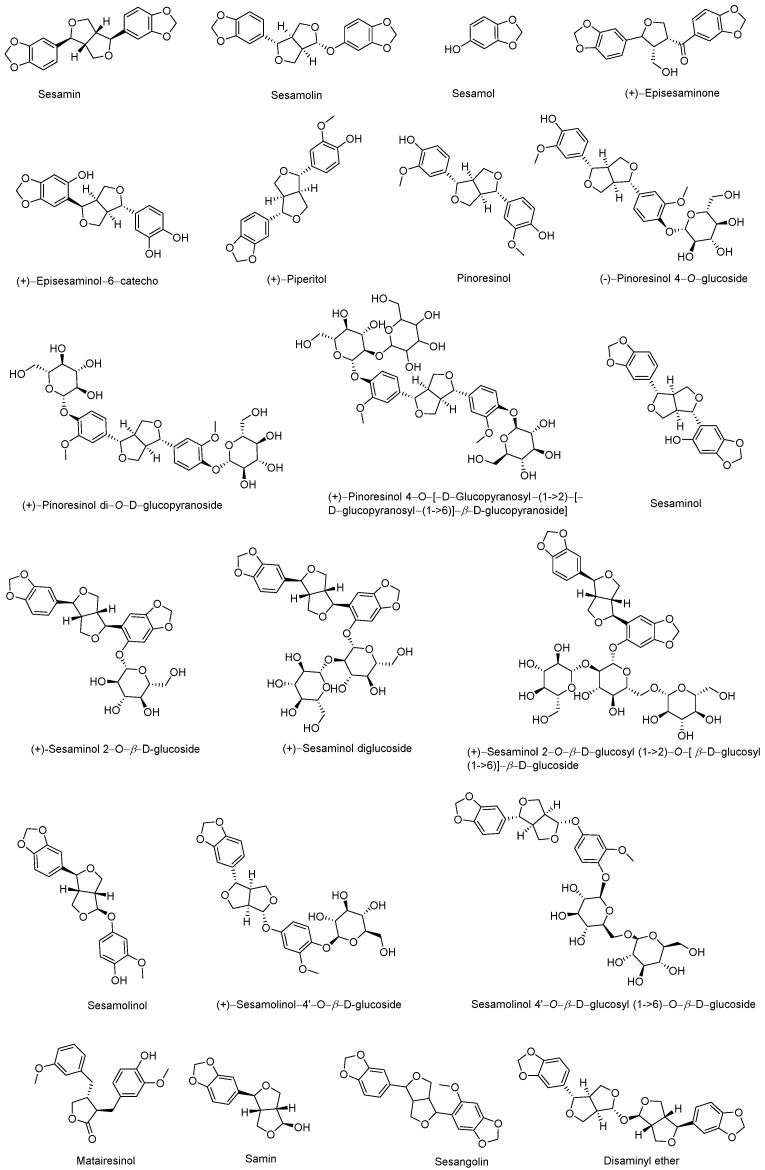
The reported chemical structures of lignans in sesame.

**Table 1 nutrients-14-04079-t001:** Main nutritional constituents of sesame.

Component	Value	Min	Max
Protein (g/100 g)	17.6	17	18
Protein, crude, N × 6.25 (g/100 g)	20.8	3.2	21.3
Carbohydrate (g/100 g)	9.85		
Fat (g/100 g)	49.7		
Sugars (g/100 g)	3	0.29	0.31
Starch (g/100 g)	4		
Fibers (g/100 g)	14.9	11.8	18
Ash (g/100 g)	4.48	4.45	4.5
Fatty acid saturated (g/100 g)	7.09	6.7	7.6
Fatty acid mono (g100 g)	18.8		18.9
Fatty acid poly (g/100 g)	21.8		21.9
Fatty acid 14:0 (g/100 g)	85	0.048	0.13
Fatty acid 16:0 (g/100 g)	4.22		4.59
Fatty acid 18:0 (g/100 g)	2.78	2.09	2.96
Fatty acid 18:1 *n*-9 cis (g/100 g)	18.7	18.6	
Fatty acid 18:2 9c,12c (*n*-6) (g/100 g)	21.2	20.9	21.5
Fatty acid 18:3 9c, 12c, 15c (*n*-3) (g/100 g)	26	0.14	0.38
Calcium (mg/100 g)	962	714	1150
Copper (mg/100 g)	1.58	1.5	4.08
Iron (mg/100 g)	14.6		
Magnesium (mg/100 g)	324	318	351
Manganese (mg/100 g)	1.24	1.17	2.46
Phosphorus (mg/100 g)	605	453	694
Potassium (mg/100 g)	468		
Selenium (µg/100 g)	26.5	2.2	51.9
Sodium (mg/100 g)	2.31	0.88	11
Zinc (mg/100 g)	5.74	5.3	7.75
*β*-Carotene (µg/100 g)	5		
Vitamin E (mg/100 g)	25		
Vitamin B1 or thiamin (mg/100 g)	79		
Vitamin B2 or riboflavin (mg/100 g)	25		
Vitamin B3 or niacin (mg/100 g)	4.52		
Vitamin B5 or pantothenic acid (mg/100 g)	5		
Vitamin B6 (mg/100 g)	79		
Vitamin B9 or folate (µg/100 g)	97		

Note: ANSES. Ciqual Table, nutritional composition of sesame seeds 2022. Source: https://ciqual.anses.fr/, accessed on 14 April 2022.

**Table 2 nutrients-14-04079-t002:** Phytochemical components in sesame.

Class of Compound	Phytochemical Components	Organ Studied	Extracting Solvent	Separation Method	Methods of Structural Verification	References
Protein	Albumin, globulin (*α* and *β*), prolamin, glutelin fractions	Seed	NA	Column chromatography	HPLC, UV	[17]
Essential amino acid	Alanine, arginine, aspartic acid, cysteine, glutamic acid, glycine, histidine, isoleucine, leucine, lysine, methionine, phenylalanine, serine, threonine, tyrosine, valine, tryptophan, proline, *γ*-aminobutyric acid	Leaf, stem, flower, seeds, root	60% MeOH	HPLC or TLC	HPLC or LC-ESI-MS/MS, HRMS, 2D NMR	[25]
Lipid	Latifonin	Flower	95% EtOH	Column chromatography	MS, NMR	[26]
Unsaturated Fatty acid	Oleic acid, linoleic acid, palmitic acid, stearic acid, arachidic acid, linolenic acid, palmitoleic acid	Seeds	EtOH	HPLC	HPLC, GC	[9]
	Lignoceric acid, caproic acid, behenic acid, myristic acid, margaric acid	Seeds	NA	Column chromatography	FT-NIR	[27]
Vitamin	Vitamin A, thiamine, riboflavin, niacin, pantothenic acid, folic acid, ascorbic acid, *α*-tocopherol, *β*-tocopherol, *γ*-tocopherol, δ-tocopherol	Seeds	NA	Column chromatography	HPLC, UV	[17]
	Tocotrienol	Seeds	MeOH or EtOH or *n*-hexane and 80% EtOH	2D-TLC	HPTLC, GC	[28]
Carbohydrates	D-Glucose, D-galactose, D-fructose, raffinose, stachyose, planteose, sesamose	Seeds	NA	Column chromatography	HPLC, UV	[17]
Lignan	Sesamin, sesamolin	Aerial organs, seeds	EtOH	HPLC	HPLC, GC	[9]
	Sesamol	Seeds	EtOH	HPLC	HPLC, GC	[9]
	(+)-Episesaminone, (+)-Episesaminol 6-catecho, pinoresinol, (−)-Pinoresinol-*O*-glucoside, (+)-Pinoresinol Di-*O*-*β*-D-glucopyranoside, glucopyranosyl-(1→6)-*β*-D-glucopyranoside, Sesaminol, (+)-Sesaminol 2-*O*-*β*-D-glucoside (+)-Sesaminol diglucoside, (+)-Sesaminol 2-*O*-*β*-D-glucosyl (1→2)-*O*-[*β*-D-glucosyl (1→6)]-*β*-D-glucoside, Sesamolinol, (+)-Sesamolinol 4′-*O*-*β*-D-Glucoside, Sesamolinol 4′-*O*-*β*-D-glucosyl (1→6)-*O*-*β*-D-glucoside, matairesinol, samin, sesangolin, disaminyl ether	Seeds	MeOH or EtOH or *n*-hexane and 80% EtOH	2D-TLC	HPTLC, GC	[28]

NA = Data not available. HRMS: high resolution mass spectrometer, NMR: nuclear magnetic resonance, TLC: thin layer chromatography, HPTLC: high performance thin layer chromatography, HPLC: high performance liquid chromatography, HR-EI-MS: high-resolution electron ionization mass spectrometry, HR-ESI-MS: high resolution electrospray ionization mass spectrometry, LC-MS: liquid chromatography–mass spectrometry, FT-NIR: Fourier transform-near infrared spectroscopy, ESI-MS: electrospray ionization mass spectrometry, MS: mass spectrometry, IR: infrared spectroscopy, GLC: gas liquid chromatography, GC: gas chromatography.

**Table 3 nutrients-14-04079-t003:** Applications of sesame in various areas.

Application	Main Findings	References
Baking Additives	Nutritional quality of the muffins was improved with addition of sesame protein isolate. Muffins with 15% sesame protein isolate are considered by professionals to have the best taste.	[32,62]
Combined with vegetable oils	Combined with other vegetable oils, sesame oil can provide balanced nutrition, anti-oxidation, and stable physicochemical properties.	[63,64]
Ghee for Baking	Chemical transesterification of sesame oil can be used to improve physical properties including SFC and melting point. The SMP and SFC decreased after interesterification. A mixture of sesame oil and palm stearin can produce trans-fat-free baking ghee.	[65]
Healthy food	The phenolic ingredients, quality characteristics, and potential beneficial properties of sesame seed oil make sesame useful in health care.	[66,67]
Antioxidants	The extract of the sesame skin in sunflower oil can enhance the oxidative stability of sunflower oil.	[67]
Massage oil	As an adjuvant treatment with sesame oil, it can effectively reduce the severity of vermic inflammation caused by chemotherapy.	[68,69,70]
Drug oil vector	Sumi is a good drug carrier that contributes to drugs through the skin barrier.	[24,50]
Feed	With the fermented sesame meal, the soybean meal in the meat improved the nutritional value of broilers, thereby increasing the production performance of broilers, which can be used as a protein source in broilers.	[71,72,73]
Fertilizer	Fermented sesame seeds are used as tobacco fertilizers, which can improve tobacco output value, enabling the quality of tobacco leaves; increase the number of bacterial, amplifier bacteria and fungi in soil; and improve the organic matter content of the soil.	[74]
Fuel	Sesame biodiesel and mixtures thereof are used to fuel engines, and in terms of fuel consumption, efficiency, and power output, they seem to have the same performance compared to mineral diesel.	[75,76]
Pesticide	Sesquiterpene has both fungicide and insecticide properties and can be used as a synergist for pyrethroid insecticides.	[77,78]
Cosmetics	The myristic acid in sesame seeds is often used as an ingredient in cosmetics.	[24,79]
Environmental Protection	Activated carbon was prepared from sesame oil cake, which was effective in the removal of hexavalent chromium from chromium plating wastewater.	[80]
Genetic modification	Diacylglycerol acyltransferase isolated from sesame was introduced into the soybean variety Dongnong 47, and Agrobacterium-mediated transformation was used to produce a soybean high-oil transgenic line more suitable for breeding.	[81]
Stabilizer for emulsions	As the amount of defatted sesame powder increases, it results in a smaller droplet size and higher stability of the emulsion against agglomeration or emulsification.	[82]

**Table 4 nutrients-14-04079-t004:** Health Functions and Related Mechanisms of Sesame.

Sample	Dose	Study Type	Experimental Model	Main Effect	Possible Compound Responsible for the Effect	Mechanism of Action	References
sesame seeds	1, 5, or 10 µM	in vitro	human neuroblastoma cell	antioxidant activity	sesamin and sesamol	ROS ↓, CAT and SOD ↑; BAX ↓; BCL-2 ↑	[102]
sesame seeds	100 mg/kg	in vivo	Cardiac hypertrophy mouse models	antioxidant activity	Sesamin	ROS ↓	[103]
sesame seeds		in vivo	male Golden Syrian hamsters	Cholesterol-lowering and lipi*D*-regulating activity	Sesamin	CYP7A1 ↑; HMG-CoA, LXR*α* ↓	[104]
sesame seeds	100 mL/kg	in vivo	male Sprague Dawley rats	protects liver and kidney activity	Sesamin	SOD, GSH-Px ↑; IL-6, COX-2 ↓; NF-κB ↓	[100]
sesame seeds	10, 20 and 40 mg/kg	in vivo	adult male Sprague Dawley rats	protects liver and kidney activity	Sesamin	ALT, AST, ALP, urea nitrogen and creatinine ↓; SOD, CAT and GPX ↑	[101]
sesame seeds	25, 50, or 100 mg/kg	in vivo	LPS-induced mouse model of AKI	protects liver and kidney activity	Sesamin	urea nitrogen and creatinine ↓; NF-κB, TLR4, COX2, TNF *α*, Nrf2 ↓	[105]
sesame seeds		in vivo	health juvenile carp	protects liver and kidney activity	Sesamin	ROS ↓; Bcl-2 ↑	[106]
aqueous extract of sesame oil	0.5, 5 and 50 ng/mL	in vitro	THP-1 monocytes, RAW 264.7 mouse macrophages	anti-inflammatory activity	methoxyphenol derivatives	LPS ↓	[107]
sesame oil		in vivo	male Wistar rats	anti-inflammatory activity	sesamol and sesamin	IL-1*β*, TNF-*α* and MCP-1 ↓	[108]
sesame seeds	0.25, 0.5 or 1 μM	in vitro	primary human synovial fibroblast cells and SW982 as synovitis models induced by TNF-*α*	anti-inflammatory activity	sesamin	TNF-*α* ↓	[109]
sesame seeds	10 and 25 μM	in vitro	mouse macrophage-like J774.1 cells	anti-inflammatory activity	sesamin and the metabolites	NO ↓	[110]
sesame seed oil		in vivo	male Sprague Dawley rats with chemically induced diabetes.	hypoglycemic activity	fat-soluble lignans, sesamin, sesamolin, and *γ*-tocopherol		[111]
sesame oil		clinical trials	type 2 diabetes mellitus patients	hypoglycemic activity	sesamin		[112]
sesame oil	0.5 or 1 mL/kg	in vivo	male specific-pathogen-free Sprague Dawley rats	protects cardiovascular system activity	sesamin and vitamin E	c-Fos, c-Jun mRNA ↓	[113]
	200 mg/day sesamin supplement	clinical trial	women with rheumatoid arthritis	protects cardiovascular system activity	sesamin		[114]
sesame seeds	50, 100 and 200 mg/kg body weight	in vivo	male Sprague Dawley rats with type 1 DM	protects cardiovascular system activity	sesamin	cardiac hypertrophy pathways are modulated by sesamin treatment.	[115]
sesame seeds	1.75–28 μM	in vitro	osteoclast	Inhibition of osteoclast differentiation	sesamin	sesamin is involved in the inhibition of ERK activation	[116]
sesame oil	100 mg/kg	in vitro and in vivo	HEI-OC1 cells, zebrafish, and noise-induced hearing loss mice	hearing protection	sesamin		[117]

## Data Availability

Not applicable.

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
