# Peer review of "Sesame (Sesamum indicum L.): A Comprehensive Review of Nutritional Value, Phytochemical Composition, Health Benefits, Development of Food, and Industrial Applications"

_nutrients, 2022, doi:10.3390/nu14194079_

Round 1

Reviewer 1 Report

Dear Authors,

The article includes an interesting analysis on the sesamum indicum plant, with information focused on aspects related in particular to the importance of this plant and its derived products.

In order to have a clearer picture of the topic addressed, I suggest some corrections or approaches to improve the presented study:

Row 17: Flax oil crops is different. In order not to create confusion, I suggest the authors to reformulate this term, so as not to be confused with linseed oil.

Row 32: For the same reason to avoid confusion - caraway is generally the term for Nigella sativa (black cumin) a related plant, but different from Sesamum Indicum, so the term caraway is not a synonym for sesame.

Row 56: For a complete description, the authors should specify the family to which this plant belongs, the genus sesamum being a member of the Pedaliaceae family.

Row 129: In the protein sub-chapter, the protein digestibility of these proteins contained in sesame should be mentioned, especially since table 3 includes the bibliographic index [52] about this parameter.

Row 168: table numbering - it is table number 2.

Subchapter 3.2: the content of essential unsaturated fatty acids is important. The authors should also include some information in a small subsection on the effects on the body of excessive use of sesame seeds (as such or processed in different forms), it being known that with excessive consumption some side effects may occur, in especially those related to blood coagulation (excessive consumption of omega 3 fatty acids can cause bleeding) or severe hypotension. Also, excessive consumption can lead to an overload of anti-nutrients.

Row 208: After the subsection "Mineral elements", I suggest the authors to insert some information related to antinutrients, nothing is mentioned about these elements contained in this plant, but in table no. 3 in the "healthy food" line mentions phenolic ingredients and it is necessary to differentiate these compounds. Sesame seeds contain natural compounds (oxalates, phenols and phytates), antinutrients that can reduce the absorption of these minerals. And also to include some methods to reduce the impact of these antinutrients on the absorption of the minerals normally contained in sesame, even if only the favorable effects of these compounds are emphasized. The only reference to antinutrients is in line 404, but in my opinion it is important to highlight a small subchapter related to these compounds.

Figure 3: the Sesaminol structure is not complete, the hydroxyl group is missing, being a 6-hydroxy-1,3-benzodioxol compound. Also, for a complete presentation of these lignans, it is necessary to add the piperitol compound, which is missing. This is an important compound in sesamin biosynthesis, as can be seen from the bibliographic index [6].

Row 375: In the research contained in the Bibliographic Index [ 57] it is briefly highlighted which compounds can contribute to the appearance of antioxidant compounds during the roasting process, as presented in the text at line 375, so the authors should add in this context some additional explanations on these compounds. It is an important quality of sesame oil that requires some additional explanations that confirm the superiority of sesame oil compared to other natural products, with reference to the shelf life.

Row 383: For more clarity, it is necessary for the authors to explain what the phrase "elevating the essence and blood" means, or to reformulate the text.

Kind regards

Reviewer 2 Report

The review entititled "Sesame (Sesamum Indicum L.): a comprehensive review of nu- 2 tritional value, phytochemical composition, health benefits, 3 development of food, and industrial applications" is very interesting providing information about sesame value with many industrial applications. Moreover, i think that it meets the publications criteria of the journal "Nutrients"

Round 2

Reviewer 1 Report

Authors complied with my suggestions for improving the article.